# Comparative Analysis of ANN-MLP, ANFIS-ACO$_R$ and MLR Modeling Approaches for Estimation of Bending Strength of Glulam

**Morteza Nazerian** [1] **, Masood Akbarzadeh** [1] **and Antonios N. Papadopoulos** [2,*]

1   Department of Bio Systems, Faculty of New Technologies and Aerospace Engineering, Shahid Beheshti University, Tehran 1983969411, Iran
2   Laboratory of Wood Chemistry and Technology, Department of Forestry and Natural Environment, International Hellenic University, GR-661 00 Drama, Greece
*   Correspondence: antpap@for.ihu.gr

**Abstract:** Multiple linear regression (MLR), adaptive network-based fuzzy inference system–ant colony optimization algorithm hybrid (ANFIS-ACO$_R$) and artificial neural network–multilayer perceptron (ANN-MLP) were tested to model the bending strength of Glulam (glue-laminated timber) manufactured with a plane tree (*Platanus orientalis* L.) wood layer adhered with different weight ratios (WR) of modified starch/urea formaldehyde (UF) adhesive containing different levels of nano-ZnO (NC) used at different levels of the press temperature (Tem) and time (Tim). According to X-ray diffraction (XRD) and stress–strain curves, some changes in the behavior of the product were seen. After selecting the best model through determining statistics such as the determination coefficient (R2) and root mean square error (RMSE), mean absolute error (MAE) and sum of squares error (SSE), the production process was optimized to obtain the highest modulus of rupture (MOR) using the Genetic Algorithm (GA) combined with MLP. It was determined that the MLP had the best accuracy in estimating the response. According to the MLP-GA hybrid, the optimum input values for obtaining the best response include: WR—49.1%, NC—3.385%, Tem—199.4 °C and Tim—19.974 min.

**Keywords:** Glulam; UF-modified starch adhesive; ZnO nano particle; MLR; ANN-MLP; ANFIS-ACO$_R$

## 1. Introduction

Glulam is an engineered wood product used for construction. The main resins used to produce these products are based on non-renewable oil resources, and although they are resistant to moisture and heat in terms of their efficiency and type of accessible material, they release toxic compounds with a low molecular mass when they are applied [1].

Today, there is a strong desire to replace synthetic resins with renewable biopolymers. Starch is one of the materials obtained from plants and is rather cheap, renewable and degradable [2]. Its bondability capacity is not as high as wood adhesive. Hence, starch must be modified to some extent using different methods [3] to meet the needs of different consumptions.

Chemical modification is one of these methods that hydrolyzes first the amorphous region and then the crystalline region, and separates the crystalline region. During this, many active sites of bonding are generated so that the bond strength improves significantly using the bond reaction [1]. The results indicate that the hydrolyzed starch adhesive has viscosity stability and connection efficiency, and is resistant to water. However, the viscosity of the starch hydrolyzed by acid for a long time does not develop in the process, so that it shows a different behavior in the grafting [4].

Different additives such as nanoparticles are used to improve the adhesion properties of wood products [5]. The extent of the bond gap and the strong stimulating binding energy have made zinc oxide important for scientific and industrial applications [6]. It was

proven that the addition of these nanoparticles improved the resistance to scratching, abrasion and corrosion in coatings [7]. When using ZnO nanoparticles with a suitable homogeneous distribution in UF resin, the increase in the pores of the resin matrix is controlled such that the physical properties of wood products bonded by UF resin are improved. In addition, these materials lead to high efficiency due to fast curing and heat transfer of resin [8]. When using nano-ZnO to produce medium-density fiberboard (MDF), it was shown that the addition of ZnO nanoparticles at 0.5% offered better results, so that it was proven that the excessive use of nanoparticles will not improve the product's properties necessarily [9]. However, in another study, it was shown that the addition of 1% ZnO nanoparticles to the particleboard increases the mechanical and physical properties. This is as a result of a better interaction between wood particles due to reaching the deformation of particles at the steady point and a better interaction, and the acceleration of resin curing [10].

When applying different ranges of press time and temperature, it was shown that at lower press times and temperatures, sufficient heat is not transferred to the board, and resin curing will not complete [11]. The results of temperature analysis show that prepolymer/starch adhesive cures at a rather low temperature, and thermal stability improves after preparation [12].

Spending less time and money, finding the relationship between variables affecting the mechanical properties such as the bending strength, and deriving predictable results with a higher accuracy, by be achieved by finding reliable methods more logically. So far, different methods such as artificial neural network (ANN), adaptive neuro-fuzzy inference system (ANFIS) and multiple linear regression (MLR) have been used in the context of wood and wood products such as plywood [13,14], laminated veneer lumber [15], oriented strand board [16,17], sandwich panel [18,19] and particleboard [20–22].

Previous methods based on neural networks and semi-empirical models are powerful tools used to model complicated phenomena, including the processes of wood composite products production, but the outputs of the methods have been different sometimes due to the difference in the accuracy when estimating the response being studied.

Hence, in the present study, the accuracy in estimating the mechanical properties of wood products is evaluated by comparing the outputs of the main modern models, including multiple linear regression (MLR), artificial neural network–multi-layer perceptron (ANN-MLP) and adaptive neuro-fuzzy inference system–ant colony optimization (ANFIS-ACO$_R$). The inputs of the models being studied are used as the independent variables, including the modified starch to UF resin weight ratio (10%, 30%, 50%, 70%, and 90%), nano-ZnO content (0%, 1%, 2%, 3%, and 4%), press temperature (120, 140, 160, 189, and 200 °C) and press time (14, 16, 18, 20, and 22 min), and the dependent variable is the modulus of rapture (MOR) of glue-laminated timber (Glulam). Due to the occasionally undesirable effects of weight application on analyses when producing different models [23], the difficulty of estimating the values of and drops in local minimum, and the decreases in the probability of detecting the optimum point and in the convergence velocity of methods [23], the Genetic algorithm (GA) was used combined with a previously developed model with the highest accuracy to estimate the response.

## 2. Materials and Methods

### 2.1. Materials

The wood used was from a plane tree (*Platanus orientalis* L.). The UF resin used was provided by Samed Mfg. & Ind. Co., Mashhad, Iran with a density of 1.26 g/cm$^3$, viscosity 350 cm, pH = 7, solid content 65% and gelation time 60 s. The corn starch was bought from Tejarat Mehravaran Chemical Company. Nano-zinc oxide was prepared by Nano Mavad Gostaran Pars Co., Tehran, Iran with the density, dimensions, purity, molecular weight and melting point of 5.6 g/cm$^3$, 20–60 nm, 97–98%, 81.38 g/mol and 1975 °C, respectively. When making Glulam, after sonicating nanoparticles according to the test design in the

aquatic medium, they were added to the adhesive and mixed to obtain a homogeneous suspension in a magnetic stirrer.

*2.2. Methods*

2.2.1. Modifying Starch and Making Starch Adhesive

The starch adhesive was made in two stages as in previous studies [24,25]. The first stage involved the chemical modification of starch; the second stage involved making the starch adhesive. First, 100 g corn starch powder was loaded in a flask containing 200 mL distilled water. After adding 10 cc NaOCl to the mixture, putting it on the heater and setting the temperature at 30 °C, it was mixed for 30 min. as its pH reached 9.5. After adding some drops of sulfuric acid 20% ($H_2SO_4$) and reducing the solution's pH to 7, then mixing for 10 min. at the temperature 30 °C, the resulting mixture was put in four 100 cc falcons. After placing these in a centrifuge at 2000 rpm for 30 min, the medium's water was separated. After washing the deposited starch with distilled water and placing it on filter paper on a Buchner funnel connected to a vacuum pump, the moisture of the mixture was removed. Due to the high concentration and the possibility that the deposited starch had coagulated, it was washed on the filter paper several times with distilled water and dried at room temperature.

In the second stage, to make the starch adhesive, after loading 50 g modified starch into a beaker and putting it inside a bain marie bath, then adding 100 mL hydrochloric acid (HCl with the concentration 2%) drop by drop to the starch and increasing the bath temperature gradually, the mixture's temperature reached 65 °C, and it was mixed for 12 min. (before reaching complete coagulation (dilatant-plastic)) [9]. After removing the mixture and measuring its pH, which was 1.5, and then adding 0.5 mol NaOH drop by drop, its pH reached 7–7.5. In this stage, the adhesion of the adhesive was observed visually and sensually. After neutralizing the pH, the mixture's temperature was set to 90–95 °C in the bain marie bath and it was mixed for 10 to 15 min. until it reached a suitable concentration. Then, the adhesive was put in an aluminum foil to dry at room temperature. Afterwards, it was ground in a ball mill, powdered, and was added to the UF resin.

2.2.2. Making the UF-OS Adhesive

According to the concentration of the adhesive used (65%), first, a certain amount of nano-ZnO was put in a beaker containing distilled water according to the test design (Tables 1 and 2), and after putting the beaker in an ultrasonic machine, a suitable distribution of nanoparticles was maintained in the aquatic medium for 30 min. Then, the prepared suspension was added to the UF resin, and after mixing manually and ensuring complete dissolution, while preparing a homogeneous solution, a certain amount of the modified starch was added to the resin according to the test design. It was mixed with a magnetic stirrer for 10 min. to ensure complete mixture within the aquatic medium and the forming of a homogeneous solution, until the adhesive was ready to use.

**Table 1.** Coded and actual values of independent variable.

| Variable | Unit | Coded Values of Variables | | | | | Actual Values of Variables | | | | |
|---|---|---|---|---|---|---|---|---|---|---|---|
| **WR** | % | | | | | | 10 | 30 | 50 | 70 | 90 |
| **NC** | % | −2 | −1 | 0 | 1 | 2 | 0 | 1 | 2 | 3 | 4 |
| **Tem** | °C | | | | | | 120 | 140 | 160 | 180 | 200 |
| **Tim** | min | | | | | | 14 | 16 | 18 | 20 | 22 |

**Table 2.** Combinations of conditions of making the test samples.

| Treatment | WR (%) | NC (%) | Tem (°C) | Tim (min.) | Treatment | WR (%) | NC (%) | Tem (°C) | Tim (min) |
|---|---|---|---|---|---|---|---|---|---|
| 1 | 30 | 1 | 180 | 16 | 40 | 30 | 1 | 140 | 16 |
| 2 | 70 | 3 | 140 | 20 | 41 | 10 | 2 | 160 | 18 |
| 3 | 30 | 3 | 140 | 20 | 42 | 50 | 2 | 160 | 18 |
| 4 | 70 | 1 | 140 | 16 | 43 | 50 | 2 | 160 | 18 |
| 5 | 50 | 4 | 160 | 18 | 44 | 50 | 2 | 160 | 22 |
| 6 | 50 | 2 | 120 | 18 | 45 | 50 | 2 | 200 | 18 |
| 7 | 30 | 1 | 180 | 20 | 46 | 70 | 3 | 180 | 20 |
| 8 | 30 | 3 | 180 | 20 | 47 | 50 | 0 | 160 | 18 |
| 9 | 50 | 0 | 160 | 18 | 48 | 70 | 1 | 140 | 16 |
| 10 | 30 | 1 | 180 | 20 | 49 | 50 | 2 | 120 | 18 |
| 11 | 90 | 2 | 160 | 18 | 50 | 70 | 1 | 180 | 20 |
| 12 | 10 | 2 | 160 | 18 | 51 | 50 | 2 | 160 | 18 |
| 13 | 30 | 1 | 140 | 20 | 52 | 70 | 1 | 140 | 20 |
| 14 | 30 | 3 | 140 | 20 | 53 | 30 | 3 | 180 | 16 |
| 15 | 50 | 2 | 160 | 14 | 54 | 50 | 2 | 120 | 18 |
| 16 | 30 | 3 | 180 | 20 | 55 | 70 | 3 | 140 | 20 |
| 17 | 30 | 1 | 180 | 20 | 56 | 70 | 1 | 180 | 16 |
| 18 | 50 | 4 | 160 | 18 | 57 | 50 | 2 | 160 | 18 |
| 19 | 90 | 2 | 160 | 18 | 58 | 70 | 3 | 180 | 20 |
| 20 | 90 | 2 | 160 | 18 | 59 | 30 | 1 | 180 | 16 |
| 21 | 70 | 3 | 180 | 16 | 60 | 30 | 3 | 140 | 16 |
| 22 | 50 | 2 | 160 | 22 | 61 | 70 | 1 | 180 | 16 |
| 23 | 30 | 3 | 180 | 16 | 62 | 30 | 1 | 140 | 20 |
| 24 | 30 | 3 | 180 | 16 | 63 | 70 | 3 | 180 | 16 |
| 25 | 10 | 2 | 160 | 18 | 64 | 30 | 3 | 180 | 20 |
| 26 | 30 | 1 | 140 | 16 | 65 | 30 | 1 | 180 | 16 |
| 27 | 70 | 1 | 140 | 20 | 66 | 50 | 2 | 160 | 14 |
| 28 | 50 | 4 | 160 | 18 | 67 | 70 | 1 | 180 | 20 |
| 29 | 70 | 1 | 180 | 16 | 68 | 50 | 2 | 200 | 18 |
| 30 | 70 | 3 | 180 | 16 | 69 | 70 | 3 | 140 | 16 |
| 31 | 30 | 3 | 140 | 20 | 70 | 70 | 1 | 180 | 20 |
| 32 | 50 | 2 | 160 | 22 | 71 | 70 | 1 | 140 | 20 |
| 33 | 70 | 3 | 140 | 16 | 72 | 50 | 2 | 200 | 18 |
| 34 | 70 | 3 | 180 | 20 | 73 | 50 | 0 | 160 | 18 |
| 35 | 70 | 3 | 140 | 16 | 74 | 70 | 1 | 140 | 16 |
| 36 | 30 | 1 | 140 | 16 | 75 | 50 | 2 | 160 | 14 |
| 37 | 50 | 2 | 160 | 18 | 76 | 70 | 3 | 140 | 20 |
| 38 | 30 | 1 | 140 | 20 | 77 | 30 | 3 | 140 | 16 |
| 39 | 50 | 2 | 160 | 18 | 78 | 30 | 3 | 140 | 16 |

### 2.2.3. XRD Analysis

X-ray diffraction (XRD) patterns were recorded for the samples of natural starch, modified starch, adhesive made of 70% starch and 30% UF resin, and 30% starch with 70% UF resin (after complete coagulation at the temperature of 160 °C) (as index adhesives) using an STOE-STADV wide-angle X-ray diffractometer (Germany) with a CuKα radiation source and a wavelength of $\lambda = 0.154$ nm.

### 2.2.4. Making Glulam

According to the EN standard, usually, 4 classes of bending strength are defined in relation to Glulam as it is one of the most well-known wood-laminated products, including $GL_{24}$, $GL_{28}$, $GL_{32}$ and $GL_{36}$, which range 24 to 36 MPa. After initial experiments, plane tree (*Platanus Orientalis* L.) was used as the most suitable wood layer to make three-layer Glulam and then determine the effects of the variables (Table 1).

For this purpose, and after cutting the tree and decreasing its moisture in the laboratory, layers with a thickness 7 mm, width 70 mm and length 350 mm were cut radially with

a band saw. Under the laboratory conditions, the boards were piled crosswise to prevent their warping and dried for two weeks. Then, the boards were put in an oven at a temperature of 140 °C for 5 h until their moisture reached 8%. Then, adhesives (150 g/m$^2$, based on the dry content) with different WR values (according to Table 2) were rubbed on. Then, the layers were pressed in the press at a certain temperature for a certain time at a pressure of 15 kg/cm$^2$. After removing the Glulam, it was air-conditioned for 72 h. After cutting the boards in such a way that the samples' width was equal to the thickness of the produced boards, the flat-wise bending test was performed in a Universal Mechanical tester with a loading speed of 5 mm/min, using the EN 310 Standard [26]. The combinations of making conditions and levels used to make the Glulam are given in Table 2.

### 2.3. Statistical Analysis

The data obtained from the test samples made based on the second-order design were derived in a factory environment, in which the number of variables at five levels was 4, and there were 24 axial and factorial points with three repetitions (72 samples) and 6 central points derived from the three separate boards in each treatment. The upper limits of the levels of every factor were coded as +2, and the lower limit was coded as −2.

Based on the factorial design used, the number of iterations at the central point of the matrix cube of the coordinate (0,0,0) was 6. The specimens were used to obtain the four main variables mentioned above by the relation $2^n + (2 \times n) + C$ with three iterations for the factorial and axial points. Hence, the total number of test specimens used was 78 (($2^4 + (2 \times 4) \times 3) + 6 = 78$) in testing the bending strength, using the Design-Expert Software Version 6 (Stat-Ease 6 Minneapolis, MN). The obtained responses were derived by applying three prediction methods, including the MLR, ANN-MLP and ANFIS-ACO$_R$, and comparing their outputs with the real values in order to choose the best modeling method.

### 2.3.1. The Multiple Linear Regression (MLR) Method

Multiple linear regression (MLR) is a statistical technique used to predict the output of a variable based on the values of one or more input variables. The variable to be predicted is known as the dependent variable, while the variables used to predict the dependent variable are known as the independent variables. MLR is based on the least squares approach. The model is developed in such a way that the square sum of the differences of the observed and estimated values is minimized. The MLR equation is given as follows:

$$Y = a_1 x_1 + a_2 x_2 + a_3 x_3 + a_4 x_4 \ldots + a_n x_n + C \tag{1}$$

where $Y$ is the dependent variable (MOR); $x_1$, $x_2$, $x_3$ and $x_4$ are independent variables (WR, NC, Tem and Tim, respectively) and $a_1$, $a_2$, $a_3$ and $a_4$ are regression coefficients, while $C$ is a constant (intercept).

### 2.3.2. The Artificial Neural Network–Multilayer Perceptron (MLP-ANN) Methods

The neural network is one of the most important branches of computational-based modeling, and is based on biological neural systems. Two types of neural networks, the radial basis function (RBF) and multilayer perceptron (MLP), are the dominant forms of ANNs. Various algorithms are used in combination with the MLP, with one common example being the back-propagation learning algorithm, which is used extensively to analyze various classifications and predict problems. The common structure of the MLP with Back Propagation comprises three input, hidden and output layers, in which the input layer is equal to the number of independent variables (WR, NC, Tem, Tim), the output layer is equal to the number of dependent variables (MOR), and the number of hidden layers is dependent on the nature of inputs and outputs and the accuracy of the response estimation, which is the most critical stage of an optimum MLP structure [27]. The neurons were determined by collecting the inputs weighed using the related bias (Equation (2)) and directing the input data toward a more nonlinear system. In this process, the number of middle-layer neurons was determined by testing the neural network to the extent that the

mean square error (MSE) of the output was minimized. When examining the effects of the independent variables on the response and evaluating the MLP's performance in predicting the response, all data were first normalized and divided into three sets, including the training (70% of all data), testing (15% of all data) and validation (15% of all data) data sets. After testing different algorithms expressed as mathematical formulae and modifying the error function to optimize the link weight, Levenberg–Marquardt Back Propagation (LM) was used. According to the outputs of the model derived when applying different forms of activation functions, the *Sigmoid* function (Equation (3)) and *tansig* function (Equation (4)) were used for the hidden layers and the *purelin* function (Equation (5)) was used for the output layer.

$$X_j = \sum_{i=1}^{p} y_i w_{ji}^{in} + b_j^{in} \tag{2}$$

where $X_j$ is the network input into node $j$ in the hidden layer, $y_i$ is the input into a neuron, $w^{in}{}_{ji}$ is the weight accompanied by each input link from the $i$-th neuron to the $j$-th neuron in the hidden layer, and $b^{in}{}_j$ is the bias of the $j$-th neuron in the hidden layer.

$$Sigmoid = \frac{1}{1 + e^{-x}} \tag{3}$$

$$Tansig = Tanh = \frac{2}{1 + e^{-2x}} - 1 \tag{4}$$

$$Linear = Purelin = x \tag{5}$$

Taking the MLP model with two hidden layers, with the *tansig* and *logsig* activation functions for the first and second hidden layers and *purelin* function for the output layer, the output can be computed as follows (Equation (6)) [28]:

$$Output = purelin(w_3 \times (Sigmoid(w_2 \times (Tansig(w_1) + b_1)) + b_2) + b_3) \tag{6}$$

where $b_1$ and $b_2$ denote the bias vectors of the first and second hidden layers and $b_3$ denotes the bias vector of the output layer. $w_1$ and $w_2$ are also the weight matrices of the first and second hidden layers, respectively, and $w_3$ is the weight matrix of the output layer.

The optimization algorithm obtained by Levenberg–Marquardt Back Propagation, which shows improved MLP performance, is shown schematically in Figure 1.

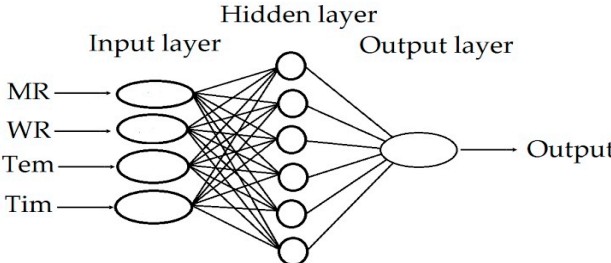

**Figure 1.** The schematic structure of the ANN-MLP algorithm used.

2.3.3. The Adaptive Neuro-Fuzzy Inference System–Ant Colony Optimization (ANFIS-ACO$_R$) Methods

The ANFIS is a feed-forward multi-layer neural network. The system uses neural network training algorithms together with fuzzy logic to develop a nonlinear design for the input–output relation. To describe the ANFIS structure, a system including two inputs ($x_1$, $x_2$), two Sugeno-type and fuzzy Takagi if–then rules, and an output (y) is normally set up as follows (Equations (7) and (8)):

$$Rule\ 1 : if\ (x_1\ is\ A_1)\ and\ (x_2\ is\ B_1)\ then\ f_1 = p_1 x_1 + q_1 x_2 + r_1 \tag{7}$$

$$Rule\ 2: if\ (x_1 is\ A_2) and\ (x_2 is\ B_2) then f_2 = p_2 x_1 + q_2 x_2 + r_2 \tag{8}$$

where *A* and *B* are fuzzy sets, and *q*, *p* and *r* are the resultant parameters of the model evaluated in the training stage.

The ANFIS structure contains 5 layers. The first layer includes the input variables of the fuzzy sets, in which all nodes adapt to a function parameter, which is normally Gaussian. In the second layer, two members multiplied by the fuzzy set are taken into account. In the third layer, each node is either fixed or non-adaptive. In the fourth layer, each node is an adaptive node with one output. In the fifth layer, there is only one node, which is either non-adaptive or fixed. This node computes the output as the sum of all incoming signals from the previous overall node.

Here, 70% and 30% of the data are used randomly for training and testing purposes, respectively. The number of epochs in the ANFIS is simple, and the number of replications in the used algorithms is 500. The optimum number of replications for all models is 500, and as this value is exceeded, no significant effects are observed on the accuracy of the model's performance. Additionally, the optimum values of the initial step-size, the step-size decrease and the initial increase are 0.01, 0.95 and 1.05, respectively. The training algorithms attempt to insert X into Equation (9) so as to minimize the difference between both sides of Equation (7).

$$\sum_{i=1}^{n} P = \sum_{i=1}^{n} \left( X \times P' \right) \tag{9}$$

Finally, the error between both sides of Equation (9) is calculated using Equation (10). If the result meets the requirements, the optimization is finished. Otherwise, the training must be performed again.

$$\sum_{i=1}^{n} E = \sum_{i=1}^{n} (V_{Target} - V_{Model}) \tag{10}$$

where $V_{Target}$ is the observed result, $V_{Model}$ is the value predicted by the model, and *E* shows the difference between the observed and predicted values. After optimization, the predicted hybrid model acts like a classical ANFIS.

Subsequently, three ANFIS methods, including the grid partition, subtracting clustering and fuzzy c-means clustering, were used to generate a basic FIS. The results indicate that fuzzy c-means clustering achieved a better performance than the others, meaning the results of the method give the performance of the classical ANFIS, which can then be applied to the $ACO_R$ algorithm.

The $ACO_R$ algorithm is a good option for the ANFIS, because it is suitable for solving complicated problems. In addition, the algorithm is never trapped in local optima. The algorithm normally tries to minimize *E* in Equation (10). After the algorithm completes its steps, other modeling steps follow, such as a simple ANFIS. In this algorithm, a continuous function, Gaussian, is used instead of a discrete function to distribute the ants inside the solution domain (Equation (11)).

$$G_{(x)}^{i} = \sum_{l=1}^{n} w_l g_l^i(x) = \sum_{l=1}^{n} w_l \frac{1}{\sigma_l^i \sqrt{2\pi}} e^{2^{\frac{(x - \mu_l^i)^2}{(\sigma_l^i)^2}}} \tag{11}$$

where $w_l$, $\sigma^i{}_l$ and *n* are the weight of the Gaussian function, the standard deviation vector, and the mean vector and effective parameter, respectively.

### 2.3.4. The Evaluation Criteria

Data were used to determine the performance of the MLR, ANFIS-$ACO_R$ and ANN methods. MATLAB (MATLAB software, version R2015a, The MathWorks, Inc., Natick, MA, USA) was used to analyze the three methods. Before the methods can produce models, the noisy data (irrelevant or too far from normal values) are cleared [29]. Accordingly, model training results in a better performance and faster convergence if it is accompanied

by data normalization, although the advantages change as the network size and sample size increase [30]. Before training the models, both input and output variables are normalized in the ranges from −1 to +1 as follows (Equation (12)):

$$X_{norm} = \frac{X_i - X_{min}}{X_{max} - X_{min}} \tag{12}$$

where $X_i$ is the input data, $X_{norm}$ is the normalized value of $X_i$, and $X_{max}$ and $X_{min}$ are the maximum and minimum data, respectively.

The accuracy of the MLR, ANN and ANFIS-ACO$_R$ models was evaluated using the determination coefficients ($R^2$) (Equation (13)), root mean square error (*RMSE*) (Equation (14)), mean absolute error (*MAE*) (Equation (15)) and sum of squares errors (*SSE*) (Equation (16)) between the measured and estimated values of the MOR parameter.

$$R^2 = 1 - \frac{\sum_{i=1}^{n}(y_i - \hat{y}_i)^2}{\sum_{i=1}^{n}(y_i - \overline{y}_i)^2} \tag{13}$$

$$RMSE = \sqrt{\frac{\sum_{i=1}^{n}(y_i - \hat{y}_i)^2}{n}} \tag{14}$$

$$MAE = \frac{\sum|y_i - \hat{y}_i|}{n} \tag{15}$$

$$SSE = \sum_{i=1}^{n}(y_i - \hat{y}_i)^2 \tag{16}$$

where $n$ is the number of observations, $\hat{y}_i$ is the predicted value, $y_i$ is the observed value and $\overline{y}_i$ is the average observations.

### 2.3.5. Combination of the ANN with the GA Algorithm

After the training process, the developed ANN model is used in additional implementations via the GA to find the optimum values of the input variables (WR, NC, Tem, and Tim), so as to maximize the MOR according to the multi-objective function and the nonlinear constraint function for the actual values in the first stage, and for the values estimated by the ANN in the second stage. The GA was treated using a four-step cycle, including the initialization of solution populations, fitness computation based on objective function, the selection of the best chromosomes, and the genetic propagation of the parent chromosomes chosen using the genetic operators, such as the crossover and mutation, to create the new population of chromosomes [31]. All processes continued until the most suitable result was obtained. The initial population was fixed at 50, the number of generations was 500, the mutation rate was 0.1 and the crossover rate was 0.85, all of which helped to achieve the best fitness. The generation process repeated for as long as the number of generations developed. When implementing the GA, the search for an optimum solution was constrained among the input ranges used in the experimental design by the nonlinear constraint function.

### 3. Results and Discussion

In Table 3, the results regarding the bending strength of Glulam made under different conditions (Table 2) are given, along with the values predicted by the models being examined, including the MLR, ANFIS-ACO$_R$ and ANN-MLP.

**Table 3.** The actual values and values estimated by the MLR, ACOR and MLP approaches.

| Treatment | Actual Value (MPa) | MLR Value (MPa) | ACO$_R$ Value (MPa) | MLP Value (MPa) | Treatment | Actual Value (MPa) | MLR Value (MPa) | ACO$_R$ Value (MPa) | MLP Value (MPa) |
|---|---|---|---|---|---|---|---|---|---|
| 1 | 94.31 | 96.89 | 86.77134 | 93.97804 | 40 | 88.48 | 80.98 | 89.13896 | 89.45788 |
| 2 | 135.75 | 112.3 | 103.4225 | 137.6192 | 41 | 97.02 | 80.34 | 125.6724 | 97.93634 |
| 3 | 105.19 | 96.64 | 104.9268 | 105.3898 | 42 | 105.08 | 91.23 | 105.7902 | 111.5015 |
| 4 | 100.16 | 117.6 | 89.13896 | 97.16088 | 43 | 98.9 | 90.45 | 106.6536 | 111.5015 |
| 5 | 116.06 | 124.5 | 103.7676 | 117.7255 | 44 | 115.29 | 101.34 | 85.90795 | 115.9854 |
| 6 | 90.2 | 105.62 | 103.4225 | 91.1306 | 45 | 82.61 | 91.34 | 104.9268 | 80.60807 |
| 7 | 90.97 | 96.64 | 124.809 | 89.23929 | 46 | 120.26 | 112.45 | 100.5366 | 118.7506 |
| 8 | 94.57 | 96.64 | 91.16155 | 92.78109 | 47 | 90.77 | 81.34 | 92.02495 | 88.25368 |
| 9 | 87.68 | 105 | 108.6761 | 88.25368 | 48 | 95.17 | 84.56 | 106.6536 | 97.16088 |
| 10 | 90.55 | 96.6 | 105.7902 | 89.23929 | 49 | 90.83 | 97.56 | 109.0212 | 91.1306 |
| 11 | 140.63 | 87.3 | 108.6761 | 139.5276 | 50 | 100.59 | 89.98 | 92.02495 | 101.2069 |
| 12 | 99.83 | 94.25 | 108.1578 | 97.93634 | 51 | 99.4 | 84.44 | 102.9042 | 111.5015 |
| 13 | 100.99 | 111.6 | 105.7902 | 104.5311 | 52 | 108.29 | 100.34 | 102.5591 | 111.4828 |
| 14 | 106.71 | 97.85 | 92.02495 | 105.3898 | 53 | 92.87 | 84.44 | 120.4188 | 102.8141 |
| 15 | 110.61 | 105.45 | 89.13896 | 109.4889 | 54 | 90.36 | 88.45 | 124.809 | 91.1306 |
| 16 | 94.21 | 90.34 | 85.90795 | 92.78109 | 55 | 140.81 | 123.34 | 122.7864 | 137.6192 |
| 17 | 86.88 | 92.34 | 106.6536 | 89.23929 | 56 | 94.89 | 88.33 | 122.7864 | 103.5971 |
| 18 | 119.81 | 111.34 | 122.4414 | 117.7255 | 57 | 100.1 | 89.99 | 119.5554 | 111.5015 |
| 19 | 140.4 | 123.44 | 111.0438 | 139.5276 | 58 | 119.18 | 132.23 | 105.7902 | 118.7506 |
| 20 | 140.54 | 124.56 | 125.6724 | 139.5276 | 59 | 93.28 | 88.88 | 122.4414 | 93.97804 |
| 21 | 131.9 | 132.45 | 111.0438 | 130.2227 | 60 | 100.16 | 88.47 | 107.8127 | 100.6346 |
| 22 | 118.2 | 104.4 | 86.77134 | 115.9854 | 61 | 112.06 | 104.56 | 102.5591 | 103.5971 |
| 23 | 114.8 | 105.55 | 88.79393 | 102.8141 | 62 | 107.65 | 101.34 | 108.6761 | 104.5311 |
| 24 | 105.05 | 113.4 | 107.8127 | 102.8141 | 63 | 136.56 | 114.56 | 85.90795 | 130.2227 |
| 25 | 100.77 | 108.45 | 108.1578 | 97.93634 | 64 | 90.99 | 81.34 | 119.5554 | 92.78109 |
| 26 | 90.15 | 95.68 | 103.7676 | 89.45788 | 65 | 93.19 | 88.94 | 102.9042 | 93.97804 |
| 27 | 112.16 | 105.67 | 120.4188 | 111.4828 | 66 | 108.63 | 104.56 | 107.8127 | 109.4889 |
| 28 | 114.9 | 109.34 | 109.0212 | 117.7255 | 67 | 100.92 | 89.99 | 109.0212 | 101.2069 |
| 29 | 105.13 | 115.34 | 122.7864 | 103.5971 | 68 | 80.05 | 77.67 | 108.1578 | 80.60807 |
| 30 | 129.54 | 112.34 | 125.6724 | 130.2227 | 69 | 129.1 | 121.34 | 91.16155 | 111.6252 |
| 31 | 109.5 | 107.56 | 122.4414 | 105.3898 | 70 | 101.78 | 98.34 | 103.4225 | 101.2069 |
| 32 | 117.09 | 104.56 | 105.7902 | 115.9854 | 71 | 115.3 | 103.45 | 100.5366 | 111.4828 |
| 33 | 95.1 | 90.76 | 86.77134 | 111.6252 | 72 | 80.17 | 87.5 | 102.9042 | 80.60807 |
| 34 | 118.18 | 114.89 | 104.9268 | 118.7506 | 73 | 85.62 | 94.56 | 105.7902 | 88.25368 |
| 35 | 97.63 | 88.57 | 88.79393 | 111.6252 | 74 | 95.29 | 100.34 | 103.7676 | 97.16088 |
| 36 | 88.92 | 93.56 | 124.809 | 89.45788 | 75 | 110.77 | 103.45 | 88.79393 | 109.4889 |
| 37 | 109.67 | 100.9 | 120.4188 | 111.5015 | 76 | 140.66 | 112.45 | 111.0438 | 137.6192 |
| 38 | 102.38 | 93.45 | 100.5366 | 104.5311 | 77 | 97.07 | 101.45 | 102.5591 | 100.6346 |
| 39 | 138.63 | 123.44 | 119.5554 | 111.5015 | 78 | 96.12 | 87.78 | 91.16155 | 100.6346 |

*Selection of the Best Modeling Method*

Several methods can be used to evaluate the effectiveness and accuracy of a certain model, and to compare two or more models, as shown in Table 4. The general predictability of a model is normally determined by $R^2$. However, the performance of a model may not be determined by $R^2$ alone. The value of $R^2$ must approach one. According to Table 4 and Figure 2, the MLP method gives the maximum $R^2$ value for the testing, training and all data sets, which are equal to 0.9105, 0.8589 and 0.8659, respectively, while the minimum values are given by ACO$_R$ and MLR, which are equal to 0.2907, 0.6065, 0.5275, 0.4759, 0.4934 and 0.4941, respectively.

**Table 4.** The statistics resulting from the testing, training and all data sets based on the MLR, $ACO_R$ and MLP approaches.

| Source | Test Data Set | | | | Training Data Set | | | | All Data Set | | | |
|---|---|---|---|---|---|---|---|---|---|---|---|---|
| | $R^2$ | *RMSE* | *MAE* | *SSE* | $R^2$ | *RMSE* | *MAE* | *SSE* | $R^2$ | *RMSE* | *MAE* | *SSE* |
| MLR | 0.2907 | 12.87 | 10.17 | 3811 | 0.6065 | 10.55 | 8.64 | 6125 | 0.5275 | 12.15 | 9.8 | 11528 |
| $ACO_R$ | 0.4759 | 10.88 | 8.80 | 2723 | 0.4934 | 11.52 | 9.12 | 7309 | 0.4941 | 19.82 | 16.09 | 30665 |
| MLP | 0.9105 | 5.16 | 3.59 | 319 | 0.8589 | 6.31 | 3.58 | 2152 | 0.8659 | 5.83 | 3.44 | 2660 |

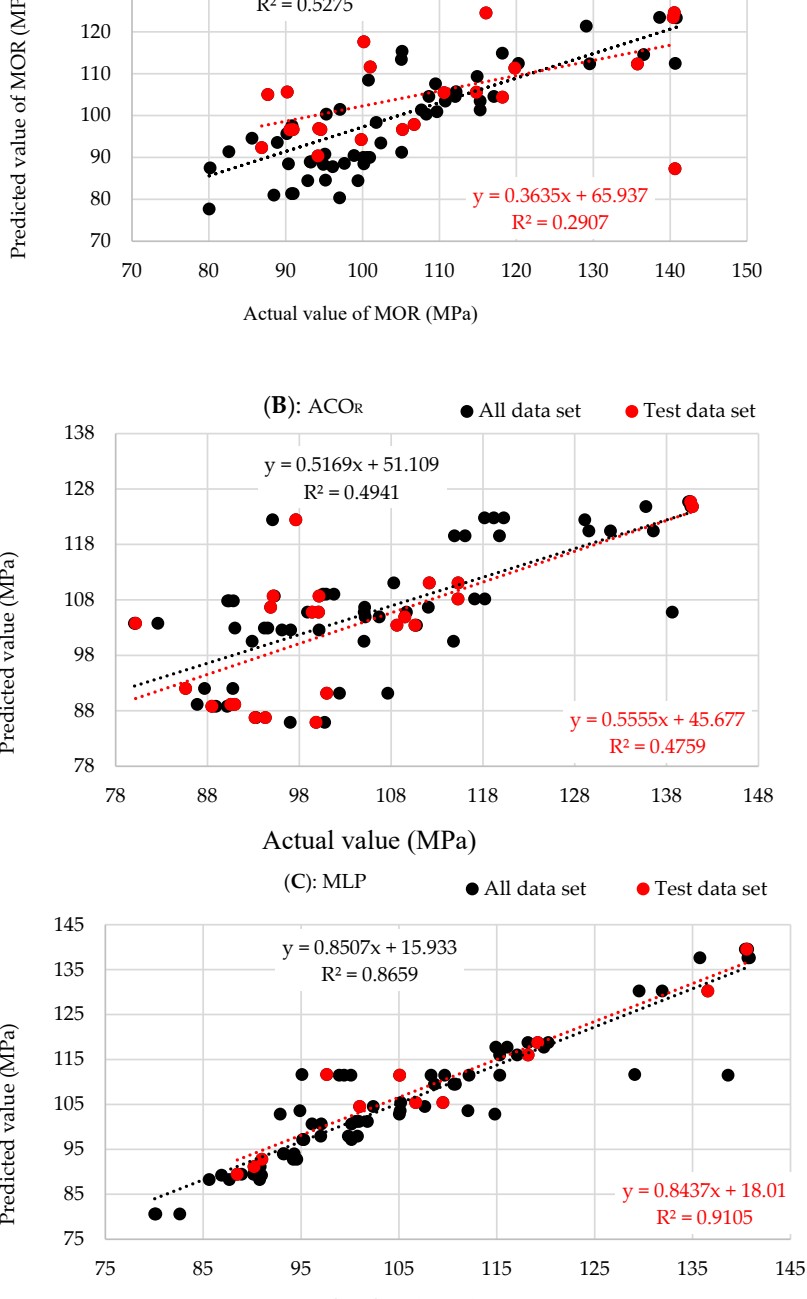

**Figure 2.** Comparison of the measured and predicted MOR (MPa) for the testing and all data sets using (**A**) MLR, (**B**) $ACO_R$ and (**C**) MLP.

The R$^2$ results of the MLP show that the developed model's data match at least 91% of the measured MOR data. Additionally, the MLR and ACO$_R$ have a lower R$_2$ value, indicating that 52.75% and 49.41% of the changes in MOR can be explained by the MLR and ACO$_R$ models. Hence, the analysis of the data given in Figure 2 shows that the MOR prediction of MLP achieves a higher accuracy compared to the MLR and ACO$_R$. The high R$^2$ values of the MLP emphasize the excellent agreement between the measured and predicted outputs of the MOR, and the high validity of the MLP in this study. Based on the MLR and ACO$_R$, we see that the models achieve a rather weak performance.

High R$^2$ values do not always mean that the regression model is efficient. Other values, such as RMSE, MAE and SSE, are also used to validate and compare more than one model. In a good model, the MAE and SSE must be as small as possible, while the RMSE must be close to zero. Higher RMSE, MAE and SSE values mean a larger probability of error in the prediction. The RMSE values are 5.16, 6.31 and 5.83, respectively, for the testing, training and all data sets. However, in MLR and ACO$_R$ modeling, the RMSE values are 2- to 3-fold greater than the values of the MLP method for all three items. The MAE values of the testing, training and all data sets obtained by MLP modeling (3.59, 3.58 and 3.44, respectively) are significantly lower than those obtained by the MLR and ACO$_R$ models. At the same time, it is observed that the SSE values of the MLP model are significantly lower (2660, 2152 and 319, respectively, for the testing, training and all data sets) than those obtained by MLR and ACO$_R$ modeling. Hence, the statistics indicate that, according to its stronger ability to predict the response, the MLP can be used for modeling and estimating the response with the greatest accuracy and reliability.

Figure 3 shows the residual error percentage of the MLR, ACO$_R$ and MLP models. It is observed that the error of the MLP model is the minimum, and the model estimating the MLR achieves the maximum error. In the MLP model, about 96.15% of the errors range from −0.01 to 0.1, while in the MLR model, 73.02% of the errors are in this range; equally, 79.48% of the errors are in this range when using the ACO$_R$, indicating the higher accuracy of the MLP model when estimating the response.

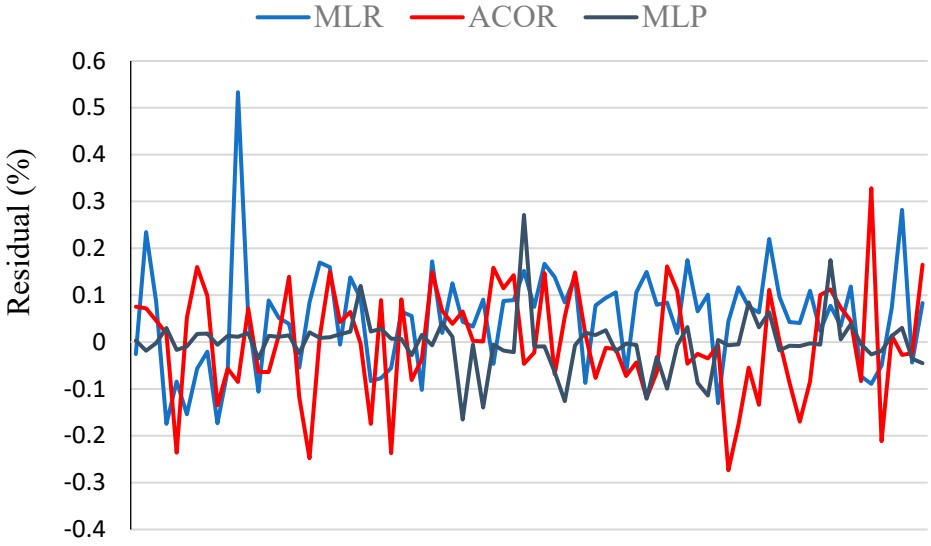

**Figure 3.** The comparison of the error values estimated by the MLR, ACO$_R$ and MLP.

The equations concerning the interaction of the variables, including $F(x_1x_2)$, $F(x_1x_3)$, $F(x_1x_4)$, $F(x_2x_3)$, $F(x_2x_4)$ and $F(x_3x_4)$, in terms of both the actual values and the values estimated by the MLP using the multiple objective function and the nonlinear constraint function, had the highest accuracy according to the statistics given in Table 5, where $x_1$ is WR, $x_2$ is NC, $x_3$ is Tem and $x_4$ is Tim. The comparison of the SSE, R$^2$, Adj.R$^2$ and RMSE values of the functions of the interactions of the variables for both sets of actual values with the values estimated by the MLP shows that the estimated values achieved lower SSE

and RMSE values, and higher $R^2$. Hence, the values obtained by combining the MLP with the GA offer a suitable estimate of the effects of every type of combination of variables on the response being examined. Using the obtained equations, the intensity of the effect of changes in each variable (including the direct, interactive and quadratic effect) can be evaluated on the response according to the (polynomial) optimized equation. It is observed that when using the equations for predicting Glulam's behavior, the regression coefficients were very close to those given by the actual equations in all estimations. Additionally, it is observed that the direct effects of the variables $x_1$, $x_2$, $x_3$ and $x_4$ (with the coefficients 2.425, 0.7444, 1.057, and 2.11, respectively), the interactive effects $x_1x_2$, $x_1x_3$, $x_1x_4$, $x_2x_3$, $x_2x_4$ and $x_3x_4$ (with the coefficients 1.766, 0.7738, 3.395, 0.4731, 2.724, and 0.8525, respectively) and the quadratic effects $x_1^2$, $x_2^2$, $x_3^2$, and $x_4^2$ (with the coefficients 3.709, 0.2929, 0.7099 and 0.5224, respectively) are different. According to the values and signs of the coefficients of every change source, we see that the interaction of $x_2x_4$ causes the greatest increase in the size of the response surface, while the greatest decrement in strength is related to $x_4$ with a negative sign. Furthermore, the function $F(x_1x_2)$ gives the best estimate in predicting the response according to its highest $R^2$ and adj.$R^2$ values (0.9444 and 0.9119, respectively) and its lowest errors (SSE = 1481 and RMSE = 4.34).

**Table 5.** The statistical criteria values of different interactions during the MLP analysis.

| Source | Function | Equation | SSE | $R^2$ | Adj. $R^2$ | RMSE |
|---|---|---|---|---|---|---|
| Actual value | $F(x_1x_2)$ | $102 + 9.018x_1 + 7.025x_2 + 4.243x_1^2 + 3.613x_1x_2 - 0.1053x_2^2$ | 8341 | 0.5785 | 0.5493 | 10.76 |
| | $F(x_1x_3)$ | $108.3 + 9.0181x_1 - 1.433x_3 + 2.932x_1^2 + 1.192x_1x_3 - 5.608x_3^2$ | 9001 | 0.4944 | 0.4593 | 11.79 |
| | $F(x_1x_4)$ | $97.87 + 9.018x_1 + 2.204x_4 + 5.099x_1^2 + 1.357x_1x_4 + 3.49x_4^2$ | 9119 | 0.4344 | 0.3952 | 12.47 |
| | $F(x_2x_3)$ | $114.8 + 7.025x_2 - 1.433x_3 - 2.774x_2^2 + 0.7229x_2x_3 - 6.967x_3^2$ | 7242 | 0.3726 | 0.329 | 13.13 |
| | $F(x_2x_4)$ | $104.4 + 7.025x_3 + 2.204x_4 - 0.6079x_3^2 - 0.3608x_3x_4 + 2.132x_4^2$ | 5454 | 0.2192 | 0.165 | 14.65 |
| | $F(x_3x_4)$ | $110.7 + -1.433x_3 + 2.204x_4 - 6.111x_3^2 - 6.386x_3x_4 + 0.8215x_4^2$ | 4204 | 0.2824 | 0.2325 | 14.04 |
| Estimated value | $F(x_1x_2)$ | $102.8 + 2.425x_1 - 0.7444x_2 + 3.709x_1^2 - 1.766x_1x_2 - 0.2929x_2^2$ | 1481 | 0.9444 | 0.9119 | 4.34 |
| | $F(x_1x_3)$ | $102.4 + 2.425x_1 - 1.057x_3 + 3.705x_1^2 - 0.7738x_1x_3 + 0.07066x_3^2$ | 1494 | 0.816 | 0.8361 | 10.38 |
| | $F(x_1x_4)$ | $103 + 2.425x_1 - 2.111x_4 + 3.654x_1^2 + 3.395x_1x_4 - 0.5224x_4^2$ | 1494 | 0.816 | 0.8035 | 11 |
| | $F(x_2x_3)$ | $108.4 - 0.7444x_2 - 1.057x_3 - 1.457x_2^2 + 4.731x_2x_3 - 1.18x_3^2$ | 1513 | 0.787 | 0.7213 | 9.5 |
| | $F(x_2x_4)$ | $109 - 0.7444x_2 - 2.111x_4 - 1.598x_2^2 - 2.724x_2x_4 - 1.773x_4^2$ | 1548 | 0.9173 | 0.9012 | 8.66 |
| | $F(x_3x_4)$ | $108.6 - 1.057x_3 - 2.111x_4 - 1.235x_3^2 - 0.8525x_3x_4 - 1.686x_4^2$ | 1584 | 0.7021 | 0.6631 | 4.83 |

Note: $x_1$: WR, $x_2$: NC, $x_3$: Tem, $x_4$: Tim.

Table 6 gives the optimum values derived after determining which MOR estimation methods offered the strongest precision based on the statistics, including RMSE, MAE, SSE and $R^2$. In this regard, using the MLP approach combined with the GA as well as the nonlinear constraint function, the optimum input values and the highest estimate of MOR can be derived according to the interactive effect of every input. It is observed that when using a WR equal to 49.1% ($-0.045$ based on the coded value), an NC equal to 3.385% (1.385 based on the coded value), a Tem equal to 199.4 °C (1.975 based on the coded value) and a Tim equal to 19.974 min. (1.987 based on the coded value), the highest MOR value can be obtained for the functions $F(x_1x_2)$, $F(x_1x_3)$, $F(x_1x_4)$, $F(x_2x_3)$, $F(x_2x_4)$ and $F(x_3x_4)$, with values equal to 110.89 MPa, 83.54 MPa, 115.54 MPa, 91.68 MPa, 124.76 MPa and 86.23 MPa, respectively. It is observed that the interactive effect of $x_2x_4$ was strongest on the response surface (function), with the highest estimate.

**Table 6.** Optimized value of the response (MOR) corresponding to each interactive effect of the variables obtained by the MLP.

| Source | Function | MOR (MPa) | $x_1$ | $x_2$ | $x_3$ | $x_4$ |
|--------|----------|-----------|-------|-------|-------|-------|
| MLP | $F(x_1 x_2)$ | 110.89 | | | | |
| | $F(x_1 x_3)$ | 83.54 | | | | |
| | $F(x_1 x_4)$ | 115.50 | $-0.045$ | 1.385 | 1.957 | 1.987 |
| | $F(x_2 x_3)$ | 91.68 | (49.1%) | (3.385%) | (199.4°C) | (19.974) |
| | $F(x_2 x_4)$ | 124.76 | | | | |
| | $F(x_3 x_4)$ | 86.23 | | | | |

Figure 4 shows the direct effects of the independent factors on the response (MOR). It is observed that as WR, NC and Tim increase, the MOR increases, while the increase in the MOR due to the increase in WR is significantly greater than the increasing effects of NC and Tim. Moreover, the effect of the press temperature increases up to a certain extent (to the middle level, i.e., 160 °C), while as the press temperature increases further beyond this, its effect on MOR decreases. It can also be observed in the figure that the experimental values of MOR in relation to the effects of each factor agree very well with the values estimated by the MLP. This agreement can be confirmed by the $R^2$ values (Figure 2) and errors (Figure 3).

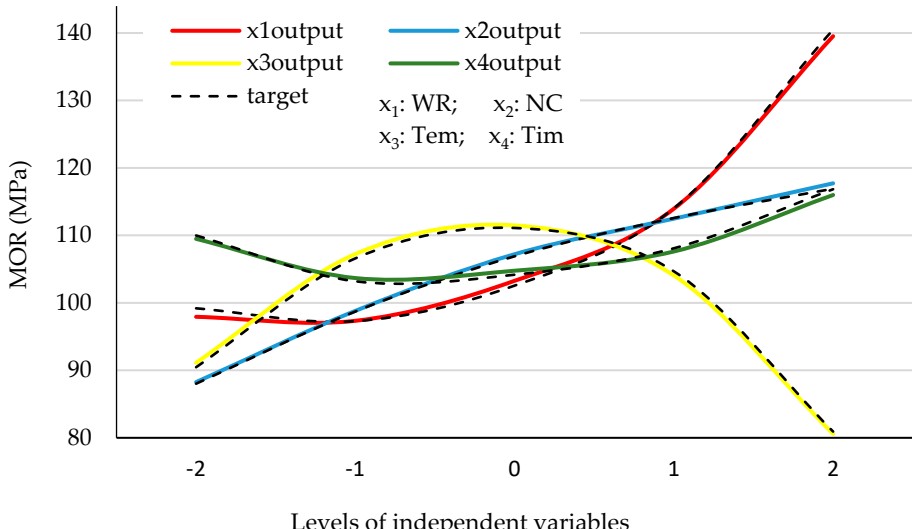

**Figure 4.** Direct effects of the independent variables on the MOR. $x_1$: WR; $x_2$: NC; $x_3$: Tem; $x_4$: Tim.

Nano-ZnO is an additive used to improve the adhesive's properties due to the small size of its particles, its high surface activity, and the unsaturated chemical bonds on its surface. Adding ZnO leads to the creation of an inter-chain network in the starch matrix formed by hydroxyl groups on the particles' surfaces. This arrangement decreases the movement of the polymer chains due to the cross-linking network created by the interaction between the starch matrix and hydroxyl groups of ZnO nanoparticles. Therefore, a more rigid structure should form, and the mechanical properties should improve. This phenomenon produces a significant effect on the moisture content of the bioplastic because the mechanical properties of starch, as a bioplastic, are largely affected by moisture [32]. The hydrogen bonds between the starch matrix and ZnO reduce the access of the hydroxyl groups in the matrix to the water molecules. As a nano-filler, ZnO creates a tortuous path for water or moisture to penetrate into the starch matrix [33,34]. Hence, the moisture around the bioplastic cannot enter into the bioplastic matrices. In addition, it is speculated that the strength increases and a good interfacial connection is created between ZnO and the starch matrix.

While the interfacial connection is an important aspect affecting the mechanical properties, the strong interaction between the starch matrix and ZnO nano-filler may be due to the homogeneous distribution of ZnO in the starch matrix. Hence, the stress transfer can be optimized and the mechanical properties can increase correspondingly. As more nano-ZnO is added and the starch increases, the viscosity is reduced, and this results in a homogeneous distribution of nanoparticles; due to the possible interactions between starch and urea and between starch and nano-ZnO, the stress can be transferred onto a larger surface, and the strength thus improves. Moreover, although adding starch increases elongation due to the plastic properties, during the bending test, it was observed that as starch content increases, the elongation increases, but at the same time, by adding more ZnO nano-filler, the elongation is controlled due to the strong interfacial interaction between the starch matrix and the ZnO nano-filler (biocomposite flexibility), such that the increase in strength continues. However, at high contents of starch and low contents of ZnO, the elongation is at its maximum.

The interactive effects of the factors on the MOR are given in Figure 5. In Figure 5a, we see that as the MR and NC increase simultaneously, the MOR reaches its maximum, such that as these two factors are reduced simultaneously, the MOR reaches its minimum. The interactive effect of WR × Tem on the MOR (Figure 5b) indicates that as WR reaches its maximum when the press temperature and NC are at their middle values (160 °C and 2%, respectively), the MOR attains its maximum. Figure 5c indicates that as WR and Tim increase simultaneously, while the other factors are held at their middle level, the MOR reaches its maximum, while the minimum MOR value is achieved when the WR and Tim are at their minimum. Figure 5d shows the interactive effect of NC × Tem on the response. It is evident that when WR and Tim are at their middle level, the NC is at its maximum and the press temperature is set at the middle level, the MOR reaches its maximum. Figure 5e shows that as NC and Tim increase and the other factors being examined are set at their middle levels, the MOR reaches its maximum. When NC and Tim reach their minimum values, the MOR does so too. Figure 5f shows the interactive effects of Tem and Tim when other factors are kept constant at their middle level. It is observed that as either of the variables increase and the other variable decreases, the MOR reaches its maximum, such that when the press time is at its maximum and the press temperature is at its minimum, the MOR reaches it maximum. There is an inverse relation between Tem and Tim.

It is shown that when low OS is added to the UF resin, the functional groups are mainly found in a typical UF resin. However, as more OS is used in the UF resin, the presence of –CH$_2$ groups, COO–R ester groups, aldehyde hydroxyl groups and ether connections increases. Due to this, the interaction between the functional groups of the modified starch and the C=O amide group in the UF resin increases [35–37].

Due to the destruction of the polymer chain and the mass loss change of nano-ZnO in the starch, the maximum decomposition of starch occurs at a higher temperature [32]. Simultaneously, ZnO has a shielding effect on the starch matrix, which reduces the mass loss rate of starch destruction [38]. Due to the inverse relationship between the therapeutic temperature and absorption of ZnO nanoparticles, the increase in ZnO absorption decreases the aeration temperature, while the total heat content increases as the concentration of ZnO nanoparticles increases simultaneously. As ZnO nanoparticles aggregate during starch mixing and nano-crystals form, both the aggregates and ZnO nanoparticles are coated by starch [34]. While polysaccharides can form a complex with bivalent metal ions due to the high number of coordinating functional groups (hydroxyl and glycoside groups) [39], the dissolved starch can improve the ZnO's stability in the aquatic medium (similar to resin) and prevent more aggregation of ZnO, which leads to the formation of ZnO nanoparticles capsulated by dissolved starch as a means of preparing ZnO nanopatricles.

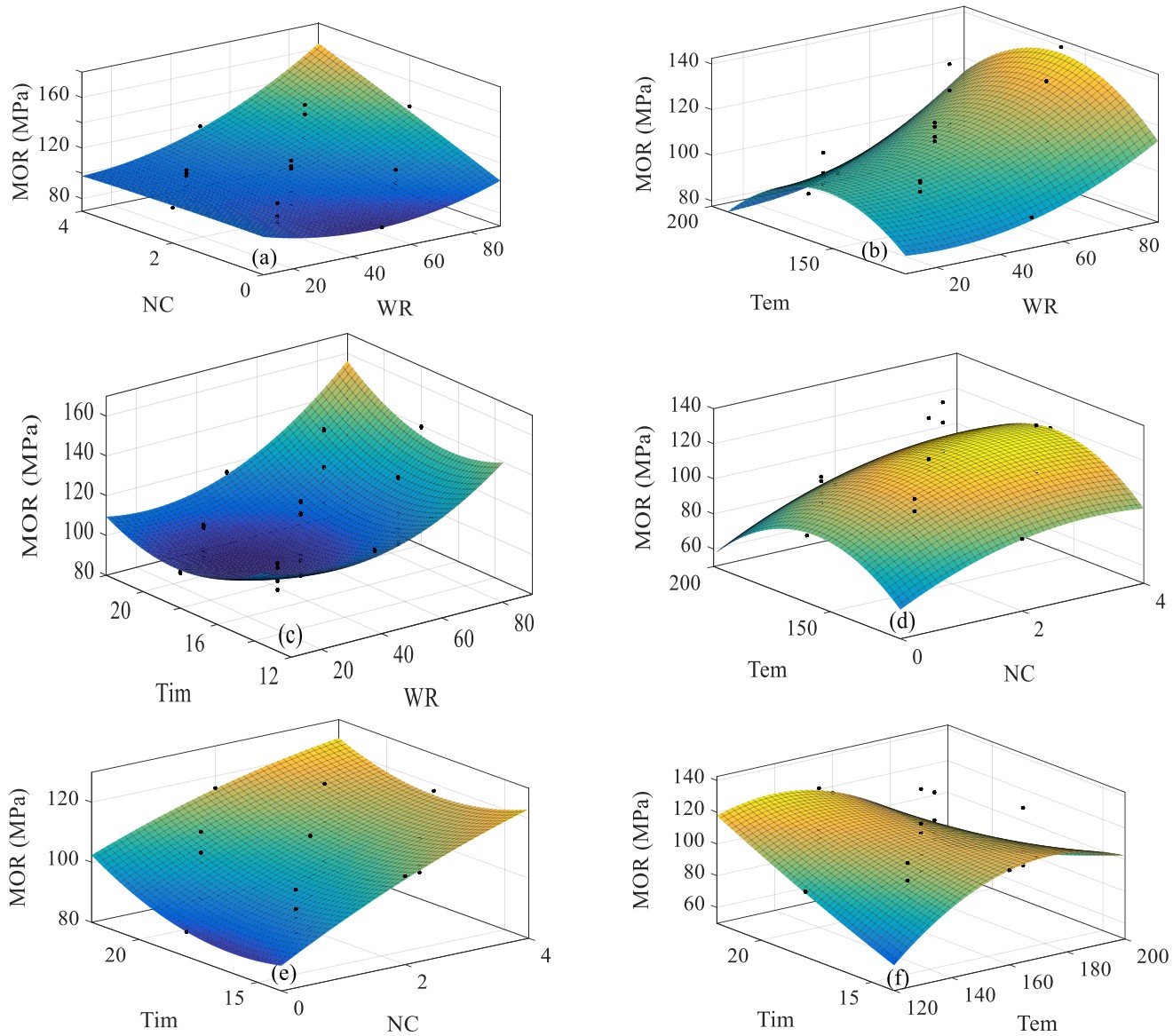

**Figure 5.** The interactive effects of WR × NC (**a**), WR × Tem (**b**), WR × Tim (**c**), NC × Tem (**d**), NC × Tim (**e**) and Tem × Tim (**f**) on the MOR.

During the complete polycondensation of the UF resin, functional groups such as –CO, –NH and –CH are generated [40]. Due to the Lewis basic nature of the NH-CO-NH group [41], urea interacts with the Lewis acid ZnO and the cross-link density increases in the CH chain. At the same time, the chemical interaction results in the stabilization of ZnO nanoparticles in the polymer matrix. Under the effect of the addition of ZnO nanoparticles, strong forces resulting from the internal bonding and Van der Waals force form between the functional groups of the coagulated resin, which leads to high thermal stability [8], i.e., the curing rate and heat transfer increase. The reaction with the UF formaldehyde leads to a heterogeneous morphology with non-uniform surfaces coated by needle-shape crystals, as determined by the U/F fraction when the agglomerated particles combine with urea with dimensions more than 200 μm. Because ZnO nanoparticles have differently oriented needle-shape crystals [42], and due to their high specific area creating active surfaces for unsaturated chemical bonds [43,44], they can be used effectively in the adhesive. They can interact with the resin matrix and convert the stress distribution simultaneously, so that the stress concentrated on the tip of the needle is transferred to other needles [37,45]. ZnO nanoparticles should be distributed homogeneously inside the resin, because otherwise,

agglomerations can increase the stress concentration and local effects [46]. Hence, needle-shaped nanoparticles can offer stronger mechanical properties when combined with the resin matrix compared to other nanoparticles with different shapes.

In UF resin, the formation of cross-links starts at 80 °C. However, at this temperature, the rate of cross-link formation is low. As the temperature increases, the rate of network formation increases [47]. This increases the shear strength. As the press temperature increases further and reaches 200 °C, the strength decreases. It was determined that a considerable mass loss occurred at 175 °C, such that beyond this temperature, the curve was descending due to starch decomposition [47]. This means that the optimum curing temperature is in the middle-range of the press temperature, according to the press time. At the same time, as the press time increases, the strength increases. A longer hot press time means a greater requirement for cross-linking energy. This shows the improvement of mechanical properties as the press time increases.

It can be seen clearly in Figure 6 that Glulams with different combinations of resin and starch, together with different levels of nanoparticles at different press times and temperatures, show different behaviors. The maximum application of starch at the middle levels of nanoparticle presence, maximum press temperature and time results in the maximum bending strength, while even in other similar conditions, such as equal levels of nanoparticles, press temperature and time, the bending strength decreases considerably as the starch content decreases. According to the stress–deflection relationship offered in Figure 6, improvements in the ductile behavior may also be a reason for this. Generally, the complete stress–strain curve model can be described as rectilinear until it reaches the failure force; then, it continues with a decline in cross-section strengthening. However, a ductility increase can be seen in the samples enriched with more starch, which is determined by the expansion of the cross-section pressure region, while in all of these components, no significant cross-section strengthening is observed after they reach the maximum force and they are exposed to a sudden decreasing destruction.

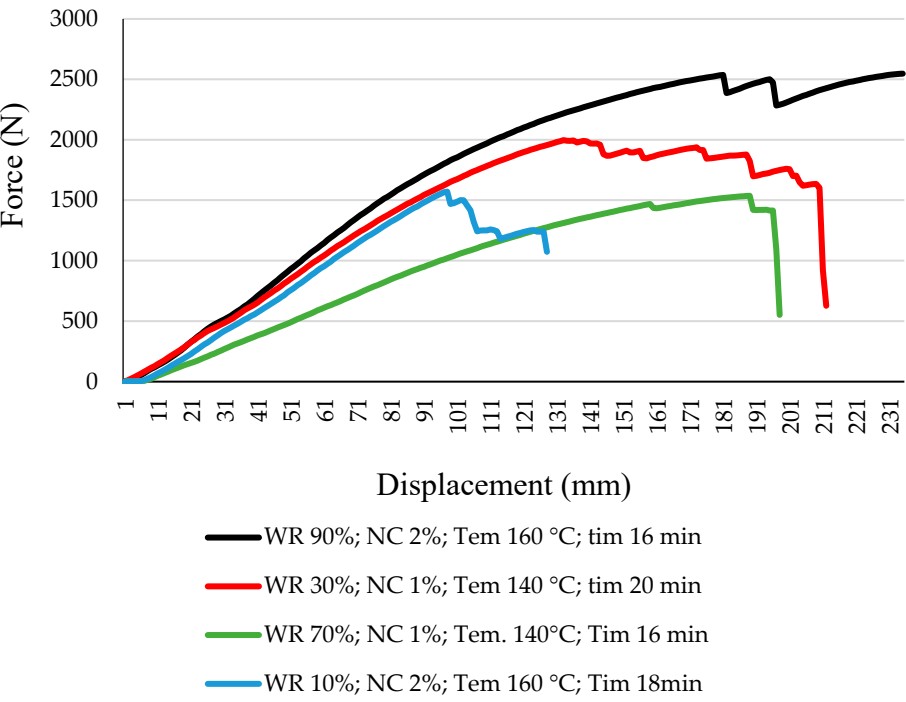

**Figure 6.** Load versus deflection curves for the selected samples.

The beam can fail under two patterns of tensile and shear stress. During the flat position test, the failure is mainly tensile. While the wood layers bear compressive or tensile stresses, the glue line layer bears the shear stress, creating delamination. In the bending

test, visual observations indicated that when using optimum values of the independent variables, the form of failure is tensile. However, as one moves away from the optimum values, the failure changes from tensile failure to the delamination mode. The reason is that the share of the glue line bending strength out of the wood layers' own strength is reduced, such that delamination mode failure mainly occurs. Therefore, the maximum bending stress does not exceed the maximum bending strength in the flatwise position, but when delamination occurs, the bending stress reaches the maximum bending strength. Hence, the shear strength causing delamination during bending, which can be distinguished based on the failure mode, offers a key insight into the determination of the optimum levels of application of the independent variables.

The XRD patterns of the natural starch and the starch modified with NaOCl are given in Figure 7. The diffractogram of the natural starch X-ray indicates the presence of peaks at 2θ = 15°, 17°, 17.5°, 19.9° and 22.8°, showing the presence of an a-type crystalline structure [48]. However, the oxidation changes the XRD pattern in starch completely.

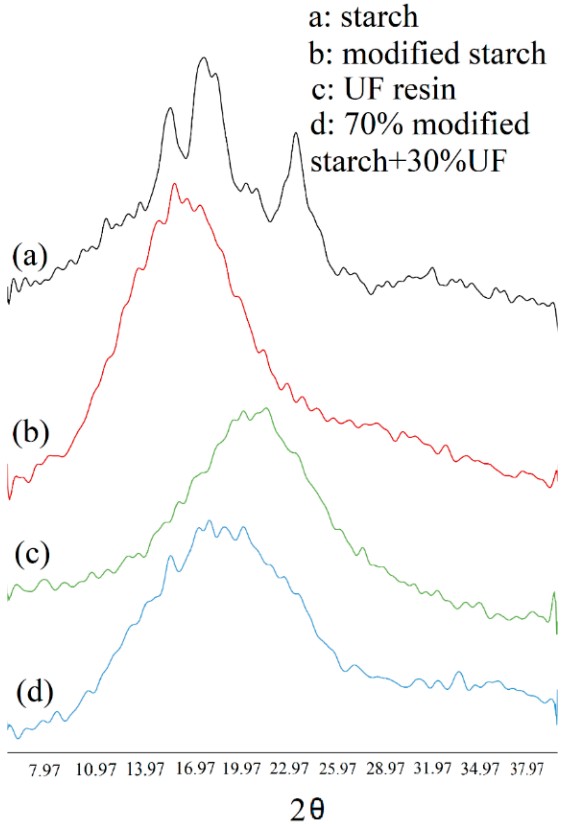

**Figure 7.** The XRD pattern for the index adhesives.

The peak at 15° is very weak, and has destroyed other peaks, showing the presence of a complete crystalline structure in the starch molecule. The steep peaks indicate that the sample contains crystallinities that are wider in the starch compared to the modified starch, and in the UF-OS compared to the UF. The shape of the general spectra in (c) is more similar to that in (d) compared to the modified starch, and especially the raw starch. The diffraction of the mixture of the modified starch and the UF resin (d) is much weaker than the modified starch (b) and the UF resin, to some extent, which may be due to the fact that the UF-OS resin has a lower concentration and viscosity, due to the breaking of the glycoside bridges in the starch in (b) and (c). In addition, the decrease in the intensity of the peak (d) may be due to the polymerization of the oxidized starch, which decreases the mobility of the resin molecules and facilitates urea branching in the resin system as a result of the modified starch. After the polycondensation reaction of the oxidized starch and urea,

the original crystalline peaks of the natural starch disappear in the U-OS completely, and strong new crystal peaks are produced instead at 37°, 37.7°, and 32°, indicating that the modified starch and urea are transferred to the U-OS adhesive.

## 4. Conclusions

In the present research, the MLR, ANFIS-ACO$_R$ and ANN-LMP methodologies were compared in terms of their ability to generalize and predict the bending strength (MOR) of Glulam made of a plane tree (*Platanus orientalis* L.) wood layer with different weight ratios of the modified starch to the UF resin, and containing different levels of nano-ZnO pressed under different time and temperature conditions. The performances of the models produced by the three methods were evaluated by different statistics. Accordingly:

- The ANN-MLP model had the best ability to offer an accurate prediction compared to the other two methods;
- After determining the ANN-MLP as the most precise method in estimating the response, and combining it with the GA, the interactive effects of the variables were derived using the multiple objective and nonlinear constraint functions, respectively, on the actual and estimated values. It was observed that the difference between the functions' factors was very slight, indicating the accurate estimation of the combined ANN-GA method used to evaluate the mechanical properties of the laminated products;
- Based on the XRD analysis, it can be observed that the chemical treatment of starch and its addition to the UF resin changes the crystallization and the chemical reactivity significantly;
- It was determined that as a result of increasing the consumption of the modified starch in the resin, together with relatively increasing the nano-ZnO in the adhesive, the behavior of the stress–strain curve improved due to the change in the ductility level.

**Author Contributions:** Conceptualization, methodology, software, validation, formal analysis, investigation and resources, project administration, writing and original draft preparation, M.N. and M.A.; investigation, visualization, writing—review and editing, writing and supervision, A.N.P. All authors have read and agreed to the published version of the manuscript.

**Funding:** This research received no external funding.

**Data Availability Statement:** The data presented in this study are available on request from the corresponding author.

**Conflicts of Interest:** The authors declare no conflict of interest.

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
