# Peer review of "Comparative Analysis of ANN-MLP, ANFIS-ACOR and MLR Modeling Approaches for Estimation of Bending Strength of Glulam"

_jcs, doi:10.3390/jcs7020057_

Round 1

Reviewer 1 Report

Overall, the quality of the paper is quite high, based on significant body of research with a sound concept and good execution. Results are important and impactful. Nevertheless, the presentation can and must be improved, as detailed below.

Overall:

- English usage is not terrible, but there are many grammar and composition problems. While this does not prevent understanding, it makes the text hard to read. 

- The use of indices is often ommited or wrong. This makes the paper hard to read. This is especially problematic in terms of parameters x1, x2, x3, and x4, (esp. in Table 4), making many of the equations and notations hard to decypher.

Materials and methods: mostly clear and easy to follow but there are some issues, as follows:

 - FTIR and TGA investigations are not described.

- Regarding the three replications mentioned in line 189, it is unclear if these three specimens came from the same plank, or three completely independent specimens. (I suspect it is the former, but it needs to be clarified.)

- In line 168, authors mention 5 levels of NC (@ 0, 1, 2, 3 and 4%), but there seems to be no 4% level in Table 1.

Results and discussion:

- In line 616: "Generally, the period of the curves can be described..." - what period? This need to be rephrased and/or clarified.

- In the first paragraph of p19 (lines 631 through 643), the authors describe tensile wood failure and shear failure (which at one point they clarify as being glueline failure). I suggest the authors use the term 'delamination' instead, as shear failure may also refer to wood failure (rolling shear).

Reviewer 2 Report

In the paper, different statistical and data analysis methodologies were compared in terms of their ability to predict the modulus of rupture of glulam made of Plane tree (Platanus orientalis L.) timber with different weight ratios of a modified starch to the UF resin containing different levels of nano- ZnO, pressed under different time and temperature conditions.

The paper is interesting but needs extensive revision before it can be published.

There are very long and complex sentences difficult to understand, which should be formulated clearer and simpler.

There is also a lot of superfluous and redundant information with respect to the purpose of the paper, which should be removed.

Data and information should be organized in tables rather than described in text.

These comments are reported in a detailed and timely manner in the attached pdf file.

Reviewer 3 Report

This is a very good study that is well written, structured, logically presented and well illustrated. I think this work will be of interest to many readers and hence will be well cited. I recommend accepting this manuscript after a minor revision.

I kindly ask the authors to shorten the Introduction and list the main goal and specific tasks of the work.

Conclusion: List Key Findings Point by Point

In general, I repeat, this is a very good and high-quality work.

Round 2

Reviewer 1 Report

Authors did address some of the problems I indicated, but I am completely puzzled by a couple of their choices:

1) Why is it so hard to use indices when naming parameters? When using x1, x2, etc., equations are very hard to read, see examples in Table 4 (remarks at highlited equations). It is also inconsistent, e.g. in eq. (1), X1 through X4 are indexed (not to mention, italicized), but in the next line (201) authors use  X1 through X4, without indexing when referencing the same parameters. Why do authors resist this so much?

2) Instead of describing FTIR and TGA methods (one or two sentences each would suffice), they eliminated mention of said tests from the text, but kept the results in. If they had not mentioned them in the first place, I would most likely not even have noticed, but now that I know they did it, what am I supposed to say? If they are going to report the results, why not just mention the tests in the Materials and methods section?

I consider this quite a remarkable and otherwise relatively well written paper, it would be a shame to keep such silly mistakes in. I hope the authors will eventually correct these.

Reviewer 2 Report

The paper has been significantly improve and is now suitable for publication after one minor revision.

I would recommend to address the comment I made in round 1 regarding point 2.1 Materials:

It would be useful to include information on the strength class of the Glulam and in particular on the values of the MOE and the MOR 

Author Response

Please see the file attached 

Reviewer 3 Report

I have no more comments. The article can be accepted in its present form.

Author Response

Thanks a lot for your comments 

Round 3

Reviewer 1 Report

I have no further notes.

Reviewer 2 Report

The paper is accepted in present form